# Detection of Low-Copy Human Virus DNA upon Prolonged Formalin Fixation

**DOI:** 10.3390/v14010133

**Published:** 2022-01-12

**Authors:** Outi I. Mielonen, Diogo Pratas, Klaus Hedman, Antti Sajantila, Maria F. Perdomo

**Affiliations:** 1Department of Virology, Helsinki University Hospital, University of Helsinki, 00290 Helsinki, Finland; outi.mielonen@helsinki.fi (O.I.M.); klaus.hedman@helsinki.fi (K.H.); 2Department of Forensic Medicine, University of Helsinki, 00290 Helsinki, Finland; antti.sajantila@helsinki.fi; 3Department of Electronics, Telecommunications and Informatics, University of Aveiro, 3810-193 Aveiro, Portugal; 4Institute of Electronics and Informatics Engineering of Aveiro, University of Aveiro, 3810-193 Aveiro, Portugal; 5Forensic Medicine Unit, Finnish Institute for Health and Welfare, 00271 Helsinki, Finland

**Keywords:** virus, DNA, formalin, nucleic acid extraction, FFPE, qPCR, NGS, hybridization capture

## Abstract

Formalin fixation, albeit an outstanding method for morphological and molecular preservation, induces DNA damage and cross-linking, which can hinder nucleic acid screening. This is of particular concern in the detection of low-abundance targets, such as persistent DNA viruses. In the present study, we evaluated the analytical sensitivity of viral detection in lung, liver, and kidney specimens from four deceased individuals. The samples were either frozen or incubated in formalin (±paraffin embedding) for up to 10 days. We tested two DNA extraction protocols for the control of efficient yields and viral detections. We used short-amplicon qPCRs (63–159 nucleotides) to detect 11 DNA viruses, as well as hybridization capture of these plus 27 additional ones, followed by deep sequencing. We observed marginally higher ratios of amplifiable DNA and scantly higher viral genoprevalences in the samples extracted with the FFPE dedicated protocol. Based on the findings in the frozen samples, most viruses were detected regardless of the extended fixation times. False-negative calls, particularly by qPCR, correlated with low levels of viral DNA (<250 copies/million cells) and longer PCR amplicons (>150 base pairs). Our data suggest that low-copy viral DNAs can be satisfactorily investigated from FFPE specimens, and encourages further examination of historical materials.

## 1. Introduction

The preservation of biological specimens through formalin fixation and paraffin embedding (FFPE) provides an invaluable resource for retrospective molecular and histological investigation. However, although this procedure confers stability to the biomolecules of a sample, it also induces DNA fragmentation, cytosine deamination, and cross-linking [1], all of which can significantly impair nucleic acid-based testing. While cross-links can be partially reversed during extraction [2], the effect on DNA integrity is permanent and directly associated with the duration of fixation, temperature, and tissue type [3].

DNA damage is of particular concern in the detection of low-frequency targets, such as persisting DNA viruses, present normally in scales of parts per million of the total genomic material in a sample [4,5,6]. In virtue of the minute amounts, polymerase chain reaction (PCR)-based methods have been the gold standard for their high sensitivity and specificity; yet, given their constraints to specific targets and intact template areas, these techniques are affected by the chemical and mechanical transformations induced by the fixation [7]. Hence, in recent years, the validation of next-generation sequencing (NGS) to FFPE samples has gained momentum for its versatility and multiplex output [8,9]. However, albeit valuable, most of these validations [8,9] lack paired comparison of fixed and non-fixed tissues from the same individuals.

Most importantly, the analytical sensitivity of detection of low-copy viral nucleic acids in archival and/or highly degraded samples has not been assessed. Indeed, most of the existing reports on FFPE tissue specimens have been PCR-based and focused on oncogenic viruses in tumor samples, in which the viral copies are commonly very high [10,11,12,13,14]. For example, in cervical cancer, human papillomaviruses can have average copies of 10^7^/µg in contrast to 10^2^/µg of most persistent viruses in healthy tissues [15]. Yet, given the clinical significance of latent viruses, assessing the sensitivity of detection and identifying optimal methods for detection in FFPE specimens is essential for the retrospective analysis of their association to specific disease states.

In the present study, we aimed at evaluating the sensitivity of detection of low-copy viral DNAs in formalin-fixed samples across prolonged fixation times. To this end, we analyzed three organs from four individuals, incubated in buffered formalin for up to 10 days (±paraffin). We compared the findings of 38 DNA viruses in the fixed samples with those of the respective frozen tissues from the same individual, using quantitative PCRs (qPCRs) and targeted enrichment plus NGS [6].

## 2. Materials and Methods

### 2.1. Sample Processing

We collected lung, liver, and kidney specimens from four individuals during post-mortem examinations. A subsection from each specimen was frozen (−20 °C) as the baseline control, while the rest of each tissue was fixed in 4% neutral buffered formalin and subsampled after 1, 2, 4, 6, 8, and 10 days of incubation at room temperature. At each collection time point, the formalin-fixed samples were either preserved in 70% ethanol (FF) or dehydrated in 70% ethanol and subsequently embedded in paraffin (FFPE). The specimens were stored at room temperature for up to 90 days until extraction. A diagram of the sample processing is presented in Figure 1.

### 2.2. DNA Extraction Methods and Quality of Total DNA

Approximately 100 milligrams of FF and frozen samples were cut into small pieces with sterile scalpels and extracted with the Qiagen QIAamp DNA Mini Kit. The FFPE specimens were sampled with 4 mm disposable sterile punches, deparaffinized using xylene/ethanol, and extracted with the Qiagen QIAamp DNA FFPE Tissue Kit (Figure 1). The DNA extractions were performed according to the manufacturer’s guidelines, except for the digestion, which was done overnight at 56 °C. The processing of the FFPE samples included an incubation at 90 °C for 60 min for formalin cross-linking reversal as recommended.

The DNAs of frozen and FF samples were eluted in 150 μL of AE buffer, while those of the FFPE samples were in 100 μL of ATE buffer, according to the manufacturer protocol. Negative controls were included along with the DNA extractions.

The quantity and fragment size of the total DNA of the frozen and FFPE samples of one individual were analyzed with a LabChip GX Instrument (Perkin-Elmer). The Genomic DNA Quality Score (GQS) given by this analysis represents the degree of degradation of a sample calculated from the size distribution, where 5 equals to intact genomic DNA (gDNA) and 0 to highly degraded gDNA.

### 2.3. Quantitative PCRs

The prevalences and copy numbers of human parvovirus B19 (B19V), human torque teno virus (TTV), and the nine human herpesviruses (HSV-1, HSV-2, VZV, EBV, HCMV, HHV-6A, HHV-6B, HHV-7, and KSHV) were analyzed by qPCRs, as previously described [6,16,17]. The lengths of the qPCR amplicons ranged between 63–159 nucleotides (nt) as follows: TTV 63 base pair (bp), B19V 154 bp, HSV-1 124 bp, HSV-2 90 bp, VZV 139 bp, EBV 90 bp, HCMV 68 bp, HHV-6A 76 bp, HHV-6B 83 bp, HHV-7 159 bp, and KSHV 159 bp.

In addition, the single-copy human RNase P gene (84 bp amplicon) was quantified as a positive control, to normalize the cell quantities [16] and for DNA integrity assessment.

Plasmid dilution series were used as positive controls and for quantification. All the qPCR reactions were performed in duplicates. The amplifications of B19V and herpesviruses were completed with AriaMx Real-Time PCR System. RNase P and TTV were analyzed with Stratagene Mx3005 P qPCR System.

Strict precautions were taken in sample handling and processing, including the use of disposable plastics and filter tips, hoods exclusively dedicated to nucleic acid work, and non-template controls throughout all steps. The reaction mixes, DNA extracts, plasmid controls, and amplification reactions, were handled each in separate rooms.

### 2.4. Library Preparation, Viral Enrichment, and DNA Sequencing

For the targeted enrichment we used the frozen and FFPE samples of one individual.

The libraries were prepared using the KAPA HyperPlus kit (Roche, Basel, Switzerland) following mechanical fragmentation with a Covaris E220 instrument with a target of 200 bp fragments. Each sample was labeled with unique Dual Index Adapters (Roche, Basel, Switzerland).

Targeted enrichment of the viral DNAs was performed using a custom panel of biotinylated RNA-baits (Arbor Biosciences, Ann Arbor, MI, USA), as described [6]. Two consecutive rounds of hybridization on individual samples were performed following the recommendations for low input DNA (MyBaits v5 kit; Arbor Biosciences, Ann Arbor, MI, USA). The baits were 100 bp in length and designed with 2X tiling. Kapa Universal Blockers (Roche, Basel, Switzerland) were used to block unspecific binding to the adapters during hybridization.

During library preparation and viral enrichment, the libraries were amplified 3 × 13-25 cycles. The cleanup steps were performed with 0.8× KAPA Pure Beads (Roche, Basel, Switzerland). The enriched libraries were quantified with KAPA Library Quantification Kit (Roche) using Stratagene 3005P qPCR System (Agilent, Santa Clara, CA, USA) and subsequently pooled for sequencing on NovaSeq 6000 (one lane; S4 PE151 kit; Illumina, San Diego, CA, USA).

### 2.5. NGS Data Analysis

The viral genomic sequences were reconstructed using alignment-based and *de novo* assembly with TRACESPipe [18]. Precisely, the adapter sequences were trimmed using Trimmomatic by removing content from an adapters’ list, with a maximum mismatch that allowed a full match of 2. The palindrome and simple clip threshold were set at 30 and 10, respectively. The minimum quality score required to keep a base at the beginning and the end was fixed to 3. Low-quality data were filtered using a sliding window of 4 with an average quality of 15, and low complexity regions were flagged with GTO [19]. Reads shorter than 20 bases were discarded. The assembly was performed using iterative refinement between alignment-based and *de novo* approaches. For the alignment approach, each highest similar reference sequence from our viral database was selected using the FALCON-meta at level 47 [20], followed by alignment of the respective reads using bowtie2 [21] with parameters for high sensitivity. Consensus sequences considering the removal of duplications were generated with SAMtools [22] and BCFTools [23]. For *de novo* assembly, metaSPAdes was used [24]. The blending between both approaches was performed using five rounds while maximizing the genomes’ size and quality. The final reconstructed sequences, as well as individual sequences when in low coverage (<15%), were manually inspected and confirmed by BLAST.

TRACESPipe code is freely available at https://github.com/viromelab/tracespipe (accessed on 1 November 2021)

### 2.6. Statistical Analysis

Non-parametric Mann–Kendall test was used for the analysis of the trend of the median RNase P ratio at the different incubation time points (Day specific RNase P yield/Day 1 RNase P yield of either FF or FFPE specimens) and the number of false-negative (FN) viral findings. FN were defined as the absence of viral detection in the fixed samples vs. a positive finding in the frozen controls. The differences in the distribution of the RNase P ratios between organs (lung, liver, and kidney of the four individuals) at the different time points was assessed by the Kruskal–Wallis test, for analysis of the variance of multiple independent samples.

After logarithmic (log_10_) transformation of the data, we used a multiple linear regression model, adjusted for organ (lung, liver, and kidney) or individual, to estimate the relationship between the virus copy numbers in the frozen controls and FNs in the FF/FFPE specimens. We also applied linear mixed models to assess the random effects of the organs or individuals independently.

The Spearman’s correlation coefficient (Rs) was calculated to estimate the strength of the linear relationship between the ordinal data (amplicon length and FN rate, and fixation time and virus copy numbers) in the FF and FFPE specimens.

The statistical analysis was performed using R, version 4.0.2. *p*-values were reported as two-sided and ≤0.05 was considered statistically significant.

## 3. Results

### 3.1. Total DNA Analysis

For control of efficient yields and viral detections, we extracted the DNA (in total 156 samples) using either a standard tissue protocol (FF and frozen samples) or a dedicated one for the processing of FFPE samples.

To compare the yields in each group, following 1 to 10 days of incubation in formalin, we quantified by qPCR the single-copy human RNase P gene. We calculated the ratios of RNase P copies of FF or FFPE samples at the different time points in relation to day one. The FFPE samples, extracted with the FFPE dedicated kit, displayed higher RNase P amplification rates compared to the FF samples, extracted with the DNA Mini Kit (Figure 2a). Upon extended formalin fixation, we observed a decreasing trend in the RNase P ratios in both groups, which was statistically significant in the FF series (FF: *p*-value 0.03, FFPE: *p*-value 0.09).

We found no substantial differences in the distribution of the RNase P ratios across organs (lung, liver, and kidney) in either the FF or FFPE specimens (*p*-value range 0.198–0.981). Amid the three tissue types, the liver showed the poorest recovery of human DNA overall, with the lowest values in four out of five time points both in the FF and FFPE groups (Figure 2b).

We also evaluated the integrity of the total DNA of the three frozen tissue controls and the corresponding FFPE samples from Day 2 and Day 10 with a LabChip GX Instrument. While the frozen controls had fragment length distributions (FLDs) close to those of intact gDNA (20–40 kb) and GQS of 3.2/5.0, the corresponding FFPE samples, fixed during 2 and 10 days, had respective median FLDs of 0.5 and 0.2 kb (GQS of 1.1/5.0 and 1.0/5.0) (Figure 3).

The kidney demonstrated the best gDNA preservation overall among all tissue types across fixation days (Figure 3).

### 3.2. Viral DNA Detection in Frozen Controls and Formalin-Fixed Samples by qPCR

We determined the prevalences of 11 different virus types in the frozen lung, liver, and kidney specimens from the four individuals, as a reference for the detection in the fixed samples.

We performed singleplex qPCRs for B19V and TTV, and three multiplex qPCRs for all the nine HHVs. The amplicon sizes ranged between 63 and 159 nt. For normalization of the viral copy numbers across tissues, we quantified the human single-copy gene RNase P.

In the frozen tissues, we found DNA positivity for five different virus types, of which HHV-6B and B19V were the most prevalent across individuals (4/4 and 3/4, respectively) and tissue types. Other viruses detected, in order of frequency, were HHV-7 (3/4), TTV (1/4), and EBV (1/4). The distributions and viral copy numbers in each of the different tissues of the individuals are presented in Table 1.

In general, the calculated copies/million cells were low, being the corresponding means for EBV 14.6, B19V 768, TTV 587, HHV-6B 990, and HHV-7 26.9.

We then compared the viral DNA findings of the frozen samples with those of the formalin-fixed (both FF and FFPE). From the total of 29 positive hits for viral DNA found in the frozen tissues of the four individuals, by day 1 the total virus positivity was 23 in both the FF and FFPE groups and moderately decreased along longer incubation times (FF: 24, 18, 17, 19, 16, and FFPE: 23, 19, 19, 18, 20 for days 2, 4, 6, 8 and 10, respectively). The increase in the number of FN calls from day 2 to day 10 was more consistent in the FF group, extracted with standard tissue kit, than in the FFPE group, extracted with dedicated FFPE protocol, although not statistically significant (FF: *p*-value 0.22 and FFPE: *p*-value 0.61). In Figure 4 the agreement of the qPCR findings is presented for the FF and FFPE samples.

Based on a multiple linear regression model, in which organs or individuals were set as fixed variables, low baseline viral copy numbers in the frozen controls correlated with higher FNs (*p*-values of FN coefficients <0.01) both in the FF (organ: adjusted R^2^ 0.383; individual: adjusted R^2^ 0.2643) and FFPE groups (organ: adjusted R^2^ 0.5146; individual: adjusted R^2^ 0.4424). This was observed in particular with EBV and HHV-7, with respective averages of 14.6 and 26.9 copies/million cells in the frozen tissues (Figure 4). According to a linear mixed model, the differences between organs explained only 11.1% of the variance in the FF and 11.0% in FFPE groups, while no variation was detected in relation to the individuals.

The rate of FNs correlated with longer amplicon lengths (FF Rs 0.50 and *p*-value of 0.0058; FFPE Rs 0.58 and *p*-value of 0.0011), being the highest for HHV-7 (amplicon length 159 bp), while the lowest for TTV (63 bp).

Similarly, for RNase P, we observed a decline in the virus copies across fixation times, which was significantly more consistent in the FF group (Rs −0.27; *p*-value 0.0003), particularly for B19V (Rs −0.62; *p*-value 1.654 × 10^−6^), compared to the FFPE (Rs −0.13; *p*-value 0.099).

### 3.3. Viral Targeted Enrichment and NGS

We carried out in-solution hybridization using RNA biotinylated oligonucleotides covering the full-length sequence of 38 viruses. Besides the viruses targeted with the respective qPCRs, our panel included 27 other virus types belonging to the *Papillomaviridae, Hepadnaviridae, Parvoviridae, Polyomaviridae, Orthopoxviridae* families.

We performed the targeted enrichment in the frozen and FFPE samples of one of the individuals. Through this approach, we confirmed the findings obtained by the qPCRs, but in addition found positive signals for BK polyomavirus, JC polyomavirus, and Merkel cell polyomavirus. Moreover, we confirmed by NGS positivity for HHV-7 in the FFPE samples that were falsely assigned as negative by qPCR (Figure 5).

In paired comparison, NGS had a sensitivity of 91.67% (95% confidence interval 73.00–98.97) while qPCR of 84.62% (95% confidence interval 65.13–95.64). From this analysis we excluded TTV, for which the overall coverages by NGS were uniquely low. This is due to the unsatisfactory representation of our capture baits of this highly diverse virus species.

We recovered the highest breadth coverages for all viruses from the frozen tissues with a median of 70% of the full genomes, while the median breadth coverage of the sequences reconstructed from the FFPE samples was 15% (Figure 6). The depth coverages in the latter were also markedly reduced.

## 4. Discussion

In the present study, we investigated the analytical sensitivity of detection of low-copy viral DNAs in soft tissues, fixed in formalin at a wide range of incubation times. To address this, we systematically compared the prevalences and quantities of persistent DNA viruses in formalin-fixed (±paraffin embedding) vs. frozen lung, liver, and kidney specimens from four deceased individuals.

First, a prerequisite for successful characterization of the low quantities of persistent viral DNAs is efficient isolation of the nucleic acids [25]. We employed two tissue-DNA extraction protocols with similar chemistries, except for the inclusion of de-cross-linking at 90 °C for 1 h in the FFPE protocol. The use of high temperatures during extraction, however, has been suggested to interfere with multiplex qPCR performance [26], while lower de-cross-linking temperatures have been shown to be beneficial in the assessment of somatic variants in cancer patients [27]. The two protocols used in this study have been previously evaluated by Bozic et al. [12] who found comparable detection rates of human papillomavirus in FFPE samples, albeit higher yields of total DNA were obtained with the FFPE kit. This was also the case in our study, where we determined higher ratios of the human single-copy gene RNase P and of viral findings in the samples extracted with the FFPE kit, although the difference to the standard protocol was not significant. The latter was also tested by Lagheden et al. [13], who, using a xylene-free approach (120 °C incubation for 20 min for de-paraffinization) and overnight lysis at 65 °C, reported on optimal DNA yields and sensitivity for HPV testing.

These results highlight the need for in-depth evaluation of different DNA extraction methods as well as de-waxing protocols (mineral oils, microwave irradiation, and alkaline heat retrieval) to maximize the recovery and sensitivity of detection [28,29]. In addition, alternative fixatives (e.g., alcohol-based) should be evaluated in prospective research to control for optimal DNA stability and integrity [29,30].

Using the LabChip analyzer, we evaluated the total-DNA integrity, as the fixation time is a known factor to influence strand damage. We observed an ample degree of fragmentation of the fixed samples that correlated with the duration of formalin incubation, as previously noted by Legrand et al. [31]. The median fragment size in the fixed tissues at day 10 was 200 bp, in contrast to the near intact length distribution of the frozen samples. The degradation was also reflected in the assessment of the RNase P gene, which despite its 84 bp amplicon demonstrated significantly reduced quantities across the incubation days.

We used short amplicon qPCRs (median target size of 90 bp) for the detection of eleven different DNA viruses, including single and double-stranded types. With the viral genoprevalences in the frozen samples as baseline controls, we found the false-negative rates to be directly correlated with longer amplicon sizes (>150 bp) and low copy numbers (<250/million cells). Nevertheless, using short-amplicon qPCRs, most viral DNA findings were correctly assigned even after 10 days of incubation in formalin. It is also important to note that, some of the false negatives may be due to stochastic variation, given the extremely low copies of our target viruses in the tissues [1].

As a consequence of damage and the fact that qPCR is strictly dependent on intact targets, we also evaluated the detection of viruses using in-solution hybridization coupled with deep sequencing. This method has shown excellent performance in the characterization of highly degraded and fragmented material [32] as well as proven efficacy in the investigation of Merkel cell polyomavirus integration from FFPE samples [33]. Our custom panel and protocol were designed to capture the full-length sequences of 38 virus types [6], including highly divergent reads expected from formalin-induced damage. We found a sensitivity of detection in our virus-enriched libraries of 91.67% compared to 84.62% of qPCR. The higher positivity noted by NGS was prominent in the detection of HHV-7, which had the longest amplicon size (159 bp) and the overall lowest copy numbers in the cohort. In addition, we detected three polyomaviruses not included in our qPCR panel. The remarkable agreement between the two approaches highlights the importance of using short amplicon PCRs (150 bp or less), as well as the need for viral enrichment before sequencing, given the poor analytical sensitivity previously reported by metagenomics [34,35].

Our results emphasize the superiority of fresh samples as starting material, as the median breadth of the assembled genomes was 70% in contrast to 15% in the fixed samples. The suboptimal genome coverage in the FFPE samples may be the result of impaired amplification due to the hindrance of the polymerase binding by cross-linking [36], as well as the read-ahead function of high-fidelity DNA polymerases (such as KAPA used here) that limits the bypassing of abasic or chemically modified sites [1].

These challenges pose obvious limitations to the characterization of full-length viral genomes, particularly in valuable archival materials, in which the validation of variants as opposed to sequence artifacts can be compromised. The latter may be partly compensated by pre-treatment of the FFPE DNA with uracil-DNA glycosylase or the use of repair strategies aiming at sequence authentication.

Our data suggest that low-copy viral DNAs can be satisfactorily investigated from FFPE samples, fixed in formalin for up to 10 days. Clearly, larger cohort sizes and longer follow-ups are granted to validate the search in archival material, particularly when considering that fixation times have not historically been standardized. However, the successful detection of ancient viral nucleic acids of measles and influenza 1918 from FFPE samples [37,38], certainly give exciting grounds for future investigations.

Our data suggests the suitability of viral DNA capture and NGS as a stand-alone procedure for the multiplex screening of FFPE samples, particularly in consideration of scanty histological material and damaged templates that may be missed by qPCR [39].

## Figures and Tables

**Figure 1 viruses-14-00133-f001:**
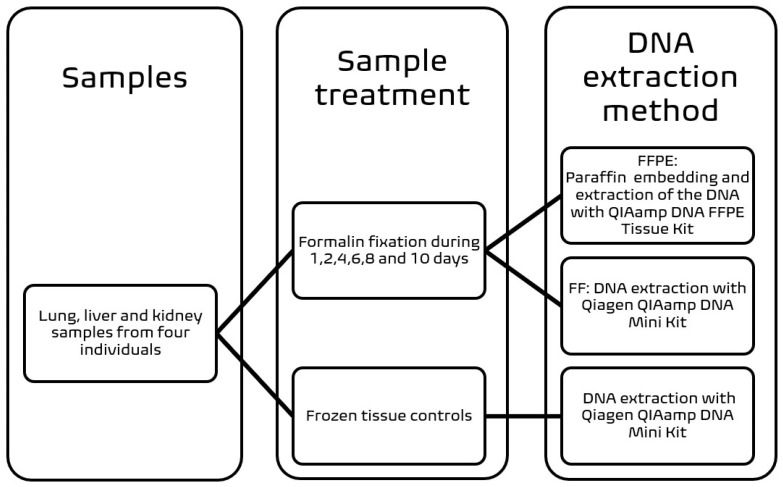
Workflow. Three tissue types from four individuals were either frozen or incubated in formalin for up to 10 days. The latter were either paraffinized or preserved in 70% ethanol until extraction.

**Figure 2 viruses-14-00133-f002:**
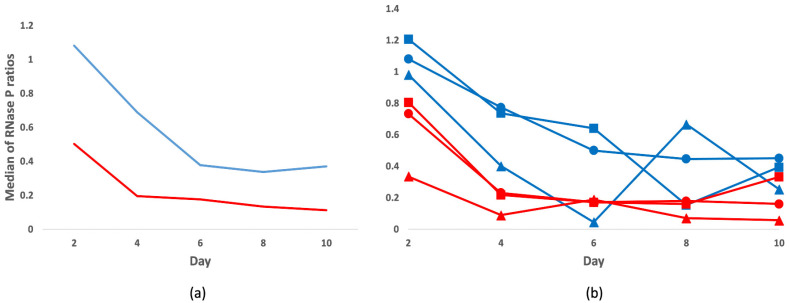
RNase P gene ratios at prolonged formalin fixation. Represented are the medians of the RNase P ratios (*y*-axis) for each time point (*x*-axis) relative to Day 1 in the FFPE (blue) or FF (red) samples. In (**a**) median ratios in each group, and (**b**) per organ [lung (circle), liver (triangle), and kidney (square)].

**Figure 3 viruses-14-00133-f003:**
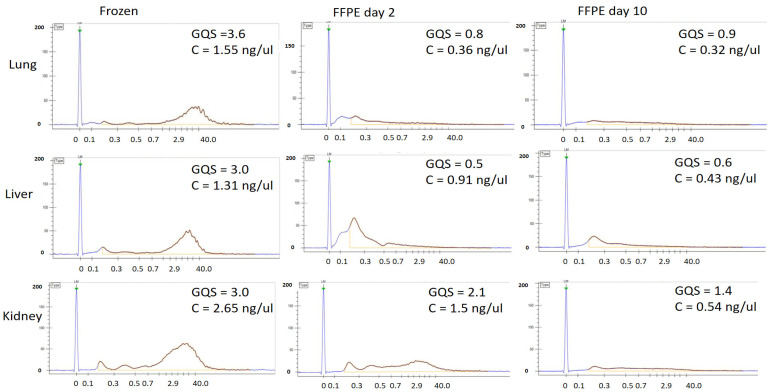
Genomic DNA integrity. Fragment length distributions and gDNA Quality Scores (GQS), of lung (top), liver (middle), and kidney (bottom) from one individual. The tissues were either frozen (left) or FFPE, following 2 (middle) or 10 (right) days of fixation. Presented is also the concentration (C) for the individual samples. *y*-axis = fluorescence intensity units, *x*-axis = fragment size (kb).

**Figure 4 viruses-14-00133-f004:**
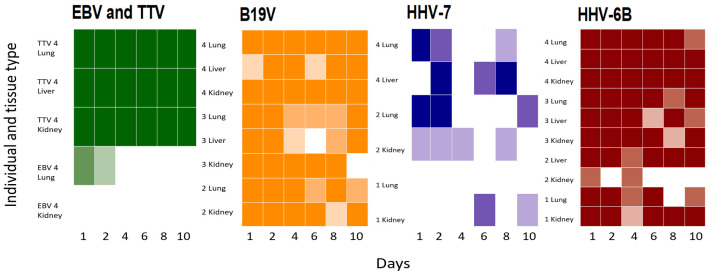
Virus DNA detection in FF and FFPE samples by qPCR. Each heat map represents the agreement of qPCR findings in the tissues of the four individuals (*y*-axis) along fixation times (*x*-axis) for the FF and FFPE samples. In dark = both FF and FFPE positive, medium = FFPE positive, light = FF positive, white = both FF and FFPE negative. From left to right and represented in colors, are EBV and TTV (green), B19V (orange), HHV-7 (purple), and HHV-6 (red).

**Figure 5 viruses-14-00133-f005:**
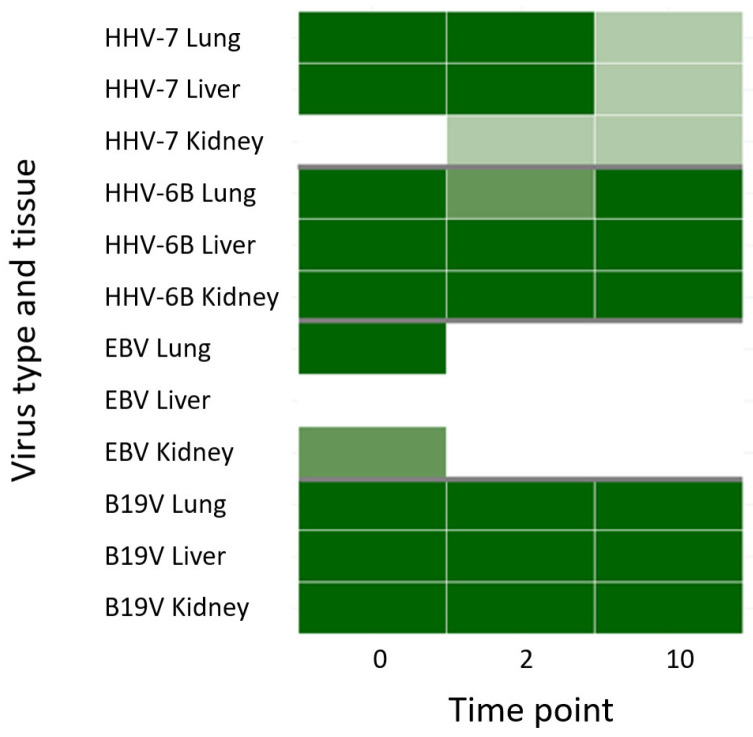
Concordance of qPCR and NGS. Represented are the positive viral findings in frozen (0) and FFPE samples of days 2 and 10 and examined by qPCR and NGS in one individual. They are in dark (both qPCR and NGS positive), medium (only qPCR positive) light (only NGS positive), and white (both qPCR and NGS negative).

**Figure 6 viruses-14-00133-f006:**
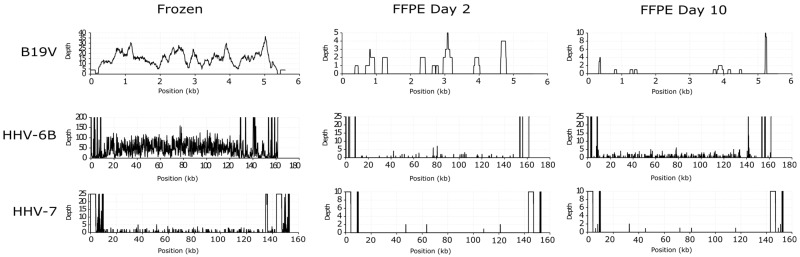
Coverage profiles of reconstructed viral DNA sequences in liver. From top to bottom, represented are the coverage profiles of parvovirus B19V (top), HHV-6B (middle), and HHV-7 (bottom) recovered from the frozen (left), and the respective FFPEs after 2 (middle) and 10 days of fixation (right). In the *x*-axis, genome position (kilobases, kb) illustrating the breadth coverage (= reads covering the noted viral reference) and in the *y*-axis, depth coverage (number of reads covering a specific nucleotide/area).

**Table 1 viruses-14-00133-t001:** Distribution and viral copies/million cells in the frozen samples.

Individual	Tissue	EBV	B19V	TTV	HHV-6B	HHV-7
1	Lung				114	27.5
	Liver					
	Kidney				165	21.6
2	Lung		472			43.6
	Liver				911	
	Kidney		101		25.4	5.79
3	Lung		691		45.7	
	Liver		50.3		249	
	Kidney		227		57.7	
4	Lung	14.6	536	154	242	23.5
	Liver		1100	1600	4720	39.4
	Kidney	14.5	2970	5.97	3370	

## Data Availability

Available upon request.

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
