# Peer review of "Detection of Low-Copy Human Virus DNA upon Prolonged Formalin Fixation"

_viruses, 2022, doi:10.3390/v14010133_

Round 1

Reviewer 1 Report

Mielonen, O.I., et al have done an exhaustive and detailed research job by inspecting the DNA of FFPE specimens obtained after different time points after formalin fixation. The study comprised 4 individuals, 3 organs each and, samples were followed up to 10 days, were authors compared both the frozen specimen, as well as the FF sample and the FFPE specimen. Both qPCR and hybridization/NGS were performed in some of the individuals for viral detection.

Major/minor comments:

  • When reading the title, “Detection of low-copy human virus DNA in formalin-fixed samples: a comparative study” , readers may have the impression that the manuscript  gathers information on archival FFPE specimens or different extraction protocols. Not until the reader reads through, he/she realizes that the study aimes for comparing different time points of formalin fixation incubation (before paraffin embedding). I believe this is something that maybe authors should consider changing or specifying, as the results from this work are so clear that operators working with FFPE material should not miss the manuscript and adapt their protocols according to the findings.
  • Why did the study stop at 10 days? Do authors know how long biopsies are formalin fixed but awaiting for paraffin embedding?
  • “Most of the existing reports on FFPE tissue specimens have been PCR based and focused on oncogenic viruses in tumor samples, in which the viral copies are expected to be high”. While this is true, there are several virologists that have both performed studies on oncogenic viruses in different parts of the body (therefore not expected to have a high viral load) and studies on all types of viruses in normal (no lesion) parts of the body. E.g: HPV and skin, metagenomics in gut. Authors have used unbiased protocols with NGS for viral detection. As authors state, NGS is more sensitive than PCR, but the existence of mutations, variants, or other viruses not targeted by primers/probes, may result in false negativity due to detection failure. This is well described in those papers, were novel or non-targeted viruses are detected with unbiased NGS and not with hybridization or qPCR.
  • Extraction method of Reference 13 (Lagheden et al), describes a comparative study of FFPE extraction methods and concludes that the method using Qiagen columns augmented with a heat treatment (xylene free method) showed better properties and was superior in all comparisons made. This method is also used by the CDC. Maybe authors would like to add that in the discussion as they are using xylene methods?
  • It is not clear if authors quantified the extracted material, and results can be affected when differences in starting material. Furthermore, elution volume was different for FFPE than for the FF/frozen. Can that correlate with the RNase P gene ratios observations?
  • Discussion lacks the part of limitations of the study (e.g. having only 4 individuals and using only 1 for different assays).
  • Discussion´s last paragraph is vague. “Our data shows the suitability of viral DNA capture and NGS as a stand-alone procedure for multiplex screening of FFPE samples, particularly in consideration of scanty histological material and the potential for highly divergent sequences that may be missed by qPCR”. Highly divergent sequences will be missed as well if not targeted by probes. I would recommend the authors to claim the main finding of their study instead.

Reviewer 2 Report

Dear Editor,

In this study, Mielonen et al. assess the effect of formalin fixation of human tissue samples on overall DNA preservation and recoverability of viral DNA. To do so, they analyze the DNA recovered from lung, liver and kidney tissues taken from four deceased individuals, each of which were either formalin-fixed or simply frozen (which they use as a control). For formalin-fixed samples, they tested different incubation time (1,2,4,6,8, or 10 days). Each formalin-fixed sample was then either preserved in 70% ethanol (FF) or dehydrated in ethanol and embedded in paraffin (FFPE; this treatment is then associated with a specific extraction method). This results in 4*3*(1 + 2*6)=156 samples in which they try to detect and quantify a set of viruses using qPCR (human parvovirus, human torque teno virus and all human herpesviruses). In addition, they quantify the human RNase P gene to normalize the results by cell quantity and assess the DNA integrity of the samples. For one individual, they further assess DNA fragment size distribution using a LabChip GX instrument. For one (other?) individual, they also assess the presence of a more diverse set of viruses using targeted enrichment with hybridization capture followed by high-throughput sequencing.

The study appears timely as formalin-fixed samples from medical collections have been recently shown to represent precious sources of ancient viral genomes. Unfortunately, the objectives are not well defined in my opinion. The authors compare a large variety of treatments with respect to different outputs without clearly stating what they want to assess and how. The statistical analyses appear very succinct, and the experimental design comes with low statistical power. It makes it difficult to grasp interesting conclusions from their results, apart from the fact that it is still possible to retrieve viral DNA in tissues after formalin fixation, which was relatively expected, I think. What is the impact of observing higher (although not significant) DNA yield and viral detection in the FFPE vs. FF samples (those are samples which were treated differently but also for which the DNA extraction protocol was different, so I am not sure to understand the implications in terms of laboratory choice)? The difference between tissue types do not seem significant (do you conclude anything form that?). What do you conclude from the observed decrease of DNA yield with incubation time? You say that your results point to higher sensitivity of the NGS based approach, but difference with the qPCR approach is not significant.

Please find below more specific comments for each section. I also attached a commented version of the manuscript with suggestions that I think should be addressed.

Introduction:

- wording is not always clear and sentences are not easy to understand. I feel that this is often due to excessive avoidance of repetition.

- please clearly state the objectives of the study at the end of the introduction

Methods:

- in the study, DNA extraction seems to be performed right after formalin-fixation: do you think this might affect the interpretation of you results for people working on old specimens that were preserved for decades before DNA extraction?

- the DNA extraction protocols used for FF/frozen and FFPE samples differ. In particular, the amount of tissue that is used cannot be compared (in one case it is ~100 µg, in the other it is a 4 mm biopsy punch). I am a bit worried that might lead to an unfair comparison, especially because 100 µg seems very little to me (the manufacturer's protocol says 25 mg of tissue if I am not mistaken)

- the NGS data analysis should be explained with more details since the procedure employed is not standard

- in my opinion, the statistical analyses should be largely rewritten. Please explain your choices and use a careful and precise language. The presentation of correlation coefficient without any idea of statistical significance seems rather limited to me. I would consider the use of mixed linear models which can allow assessing the effect of various factors on a continuous variable while accounting for the dependency of observations (i.e. the "individual" and "specimen" effects).

Results and Discussion: see comments in the attached pdf.

Round 2

Reviewer 2 Report

Dear Editor,

I am satisfied with the revisions provided. My only last suggestion would be that the authors modify the last sentence of their abstract to present the conclusions of their study in a more comprehensive way.

All the best

Author Response

We thank the reviewer for this comment. We have now modified the last sentence while complying with the instruction of a 200-word limit.